# Learning to Take a Break: Sustainable Optimization of Long-Term User Engagement

**Eden Saig**
Computer Science Department
Technion – Israel Institute of Technology
Haifa, Israel
edens@cs.technion.ac.il

**Nir Rosenfeld**
Computer Science Department
Technion – Israel Institute of Technology
Haifa, Israel
nirr@cs.technion.ac.il

## Abstract

Optimizing user engagement is a key goal for modern recommendation systems, but blindly pushing users towards increased consumption risks burn-out, churn, or even addictive habits. To promote digital well-being, most platforms now offer a service that periodically prompts users to take a break. These, however, must be set up manually, and so may be suboptimal for both users and the system. In this paper, we propose a framework for optimizing long-term engagement by learning individualized breaking policies. Using Lotka-Volterra dynamics, we model users as acting based on two balancing latent states: *drive*, and *interest*—which must be conserved. We then give an efficient learning algorithm, provide theoretical guarantees, and empirically evaluate its performance on semi-synthetic data.

## 1 Introduction

As consumers of content, we have come to rely extensively on algorithmic recommendations. This has made the task of recommending—in a relevant, timely, and personalized manner—key to the success of modern media platforms. Most commercial systems are built with the primary aim of optimizing user *engagement*, a process in which machine learning plays a central role. But alongside their many successes, recommendation systems have also been scrutinized for heedlessly driving users towards excessive and often undesired levels of consumption. This has raised awareness as to the need for redesigning recommendation systems in ways that actively promote digital well-being.

How can media platforms balance business goals with user well-being? One prominent approach, which is now offered by most major platforms, is to periodically prompt users to take breaks (Constine, 2018; Perez, 2018). The idea behind breaks is that occasional disruptions curb the inertial urge for perpetual consumption, and can therefore aid in reducing 'mindless scrolling' (Rauch, 2018), or even addiction (Montag et al., 2018; Ding et al., 2016). As a general means for promoting well-being, breaking is psychologically well-grounded (e.g., Danziger et al., 2011; Sievertsen et al., 2016). But for platforms, breaks serve a utilitarian purpose: their goal is to foster *long-term engagement* by compensating for the myopic nature of conventional recommendation algorithms, which are typically trained to optimize *immediate engagement*. Since breaking schedules are applied heuristically on top of existing recommendation policies—and typically need to be set up manually by users—current solutions unlikely utilize their full potential (Monge Roffarello and De Russis, 2019).

In this paper, we propose a disciplined learning framework for responsible and sustainable optimization of long-term user engagement by controlling breaks. Our point of departure is that *sustained engagement necessitates sustained user well-being*, and here we advocate for breaks as a means to establish both. Focusing on feed-based recommendation, our framework optimizes long-term engagement by learning an optimal *breaking policy* that prescribes individualized breaking schedules.

2022 Trustworthy and Socially Responsible Machine Learning (TSRML 2022) co-located with NeurIPS 2022.

The challenge in learning to break is that the effects of recommendations on users can slowly accumulate over time, deeming as ineffectual policies that rely on clear signs of over-exposure. To be preemptive, we argue that breaks must be scheduled in a way that anticipates the future trajectory of user behavior, and early on. To achieve this, we introduce a novel class of behavioral models based on Lokta-Volterra (LV) dynamical systems (Lotka, 1910). These depict users as acting based on two balancing forces: *drive to consume* and *intrinsic interest*, with corresponding latent states. Intuitively, high interest increases drive to consume, but prolonged consumption decreases interest; together, these describe how user behavior varies over time and in response to recommendations. Our model captures the notion that interest can exhaust long before over-consumption is observed. This arms our approach with the prescience needed to prevent burn-out by ensuring that interest is sustainably preserved; thus, whereas current solutions target the symptom—ours aims for the cause.

Our proposed learning algorithm consists of two steps: First, we embed user interaction sequences in 'LV-space'—the set of all possible trajectories that our behavioral model class can express. Then, we optimize individualized breaking policies by solving an optimal control problem over this latent space, in which the control variable is a breaking schedule applied on top of the existing recommendation scheme. Here the challenge is that different breaking policies can lead to different counterfactual trajectories, of which observational data is only partially informative. Since our goal considers long-term outcomes, our solution is to optimize directly for *counterfactual steady-states*. From a behavioral perspective, we view this as aiming to steer towards sustainable habits; from a computational perspective, under our choice of policy class, this enables tractable learning.

As we show, the optimization landscape of LV equilibria admits a compact representation, whose main benefit is that it can be fully described by *predictions* of individualized user engagement rates. Practically, this is advantageous, as it circumvents the need to take arbitrary and costly exploration steps, and enables learning using readily available predictive tools (e.g., Gupta et al., 2006). We make use of a small set of learned predictive models, each trained on a small and minimally-invasive experimental dataset, which allow us to tune our policy to suit different conditions. The final learned policy has an intuitive interpretation: it takes as input a small set of predictions for a user, and via careful interpolation, applies a decision rule that anticipates the effects of breaking on future outcomes (c.f. conventional approaches, which take in predictions and apply the myopic argmax rule).

Our main theoretical result is a bound on the expected long-term engagement of our learned breaking policy, relative to the optimal policy in the class. We show that the gap decomposes into three distinct additive terms: (i) predictive error, (ii) modeling error (i.e., embedding distortion), and (iii) variance around the (theoretical) steady state. These provide an intuitive interpretation of the bound, as well as means to understand the effects of different modeling choices. Our proof technique relies on carefully weaving LV equilibrium analysis within conventional concentration bounds for learning.

Finally, we provide an empirical evaluation of our approach on semi-synthetic data. Using the MoiveLens 1M dataset, we generate discrete time-series data in a way that captures the essence of our behavioral model, but is different from the actual continuous-time dynamics we optimize over. Results show that despite this gap, our approach improves significantly over myopic baselines, and often closely matches an optimal oracle. We also study the role of experimental treatments, and analyze how different users benefit differently from breaks. Taken together, these demonstrate the potential utility of our approach. Code is available at: https://github.com/edensaig/take-a-break.

**Broader perspective.** At a high level, our work argues for viewing recommendation as a task of *sustainable resource management*. As other cognitive tasks, engaging with digital content requires the availability of certain cognitive resources—attentional, executive, or emotional. These resources are inherently *limited*, and prolonged engagement depletes them (Kahneman, 1973; Muraven and Baumeister, 2000); this, in turn, can reduce the capacity of key cognitive processes (e.g., perception, attention, memory, self-control, and decision-making), and in the extreme—cause ego depletion (Baumeister et al., 1998) or cognitive fatigue (Mullette-Gillman et al., 2015). As a means to allow resources to replenish, 'mental breaks' have been shown to be highly effective (Bergum and Lehr, 1962; Hennfng et al., 1989; Gilboa et al., 2008; Ross et al., 2014; Helton and Russell, 2017).

Nevertheless, traditional approaches to recommendation remain agnostic to the idea that recommending takes a cognitive toll: they simply recommend at each point in time the item predicted to be most engaging (Robertson, 1977). As an alternative, our approach explicitly models recommendation as a process which draws on these resources, and therefore—must also conserve them. The subclass

of 'Predator-Prey' LV dynamics which we draw on are used extensively in ecology for modeling the dynamics of interacting populations, and demonstrate how over-predation can ultimately lead to self-extinction by eliminating the prey population—but also show how enabling resources to naturally replenish ensures sustainable relations. As such, here we advocate for studying recommendation systems as human-centric *ecosystems*, and take one step towards their sustainable design.

## 1.1 Related work

**User dynamics: latent states and feedback.** A recent body of work aims to capture time-varying behavior by modeling users as acting based on dynamic latent states. Broadly, works in this field model the effects of recommendations as either shifting user preferences via positive-only feedback (Jiang et al., 2019; Kalimeris et al., 2021; Sanna Passino et al., 2021; Dean and Morgenstern, 2022), or reducing willingness to consume via negative-only feedback, e.g. via boredom, satiation, or fatigue (Wang and Lin, 2003; Warlop et al., 2018; Kleinberg and Immorlica, 2018; Cao et al., 2020; Leqi et al., 2021). While these restrict behavior to expressing a unidirectional effect, our approach integrates both types of feedback and models internal states as competing but balancing forces, which we argue is more realistic. This draws connections to recent attempts of injecting psychological modeling into recommendation system design (Kleinberg et al., 2022; Dubey et al., 2022; Curmei et al., 2022). Works in this field often combine theoretical analysis with simulation studies (Schmit and Riquelme, 2018; Chaney et al., 2018; Mansoury et al., 2020; Krauth et al., 2020), and here we follow suit.

**Lotka-Volterra dynamics: modeling, learning, and control.** The study of ecosystem dynamics and their conservation has a long and rich history, in which LV analysis plays an integral role (see e.g. Hofbauer et al. (1998); Takeuchi (1996)). LV systems our used primarily for modeling biological ecosystems, but are also used in economics (Weibull, 1997; Samuelson, 1998), finance (Farmer, 2002; Scholl et al., 2021), and behavioral modeling (e.g., drug addiction and relapse (Duncan et al., 2019)). In terms of learning, Gorbach et al. (2017) and Ryder et al. (2018) propose variational techniques for dynamical systems, but do not consider control. Our work aims to directly learn optimal policies, for which we draw on recent advances in turnpike optimal control (Trélat and Zuazua, 2015).

## 2 Learning setting

We consider a sequential recommendation setting in which users interact with a stream of recommended items over time. New users $u$ are sampled independently from some unknown distribution $D$, and time for each user is measured relative to their time of joining. Interactions occur at discrete time-points in continuous time, $t \in \mathbb{R}_+$, and upon user request: when a user $u$ queries the system for additional content at time $t$, the system responds by presenting a recommended item $x(t) \in \mathcal{X}$. We assume recommendations are governed by an existing and fixed *recommendation policy* $\psi$, which determines $x(t)$ given a request from $u$ at time $t$. We allow $\psi$ to be stochastic, and make no additional assumptions on its structure or mechanics. User interactions are therefore described by a sequence of pairs $\{(t_i, x_i)\}_i$, where $x_i = x(t_i) \sim \psi(u; t_i) \in \mathcal{X}$ is the item recommended to $u$ at time $t_i$.

Our key modeling assumption is that subsequent request times, $t_{i+1} = t_i + \Delta t_i$ for $\Delta t_i > 0$, are determined jointly by the user's 'state' at time $t_i$ and the recommended item $x_i$ (note this means $\Delta t_i$ can depend on $t_i$). We consider user states as latent and in the abstract, but broadly expect 'good' recommendations to entail frequent requests by inducing small values of $\Delta t_i$. Together, user $u$'s choice behavior, coupled with the policy $\psi$, induce a temporal point process (TPP) which governs the generation of interaction sequences of duration $t$ as $\{(t_i, x_i) \mid t_i \in [0, t]\} \sim \text{TPP}_\psi(u; t)$.

For each user, the system collects data until some (relative) fixed time $T_0$, and seeks to optimize engagement in the subsequent interval $[T_0, T_0 + T]$, where $T$ is a predetermined time horizon. We denote the corresponding input sequences by $S_u^0 = \{(t_i, x_i) \mid t_i \in [0, T_0)\}$, and target sequences by $S_u = \{(t_i, x_i) \mid t_i \in [T_0, T]\}$. Defining by $\frac{1}{T}|S_u|$ the *engagement rate* of $u$ for the chosen time horizon $T$, our goal in learning will be to maximize expected *long-term user engagement rate*, namely $\mathbb{E}_{u \sim D} \mathbb{E}\left[\frac{1}{T}|S_u|\right]$ for $(S_u^0, S_u) \sim \text{TPP}_\psi(u; T_0 + T)$, in expectation over new users.

**Breaking policies.** The unique aspect of our learning problem is that our only means for increasing engagement is by suggesting *breaks*. Our task will be to learn a *breaking policy* $\pi \in \Pi$ which

can override $\psi$: when user $u$ queries the system at time $t_i$, the policy $\pi(u; t_i) \in \{0, 1\}$ determines whether to present the intended item $x(t) \sim \psi(u; t_i)$ (for $\pi = 0$), or suggest a break instead ($\pi = 1$).[1] Denoting the overall composed policy by $\pi \circ \psi$, our aim is to learn a breaking policy $\pi$ that increases engagement by complementing an existing $\psi$. Hence, our learning objective is:

$$\arg\max_{\pi \in \Pi} \mathbb{E}_{u \sim D} \, \mathbb{E}_{\text{TPP}} \big[ \tfrac{1}{T} |S_u| \big], \qquad (S_u^0, S_u) \sim \text{TPP}_{\pi \circ \psi}(u; T_0 + T) \tag{1}$$

Broadly, we expect breaks to negatively affect short-term engagement (i.e., entail longer ensuing $\Delta t_i$), but have the potential to improve engagement in the long run if scheduled appropriately.

**Data and exploration.** For learning an optimal breaking policy, we assume the system has access to a dataset of previously logged interactions $\mathcal{S} = \{(S_1^0, S_1), \ldots, (S_m^0, S_m)\}$ for $m$ train-time users $u_1, \ldots, u_m \sim D$, collected under a 'clean' recommendation policy $\psi$. It will be convenient to assume that the system 'featurizes' user input sequences $S_u^0$ via a vector mapping $\phi$ used in learning. For clarity, we overload notation and represent users as $u = \phi(S_u^0) \in \mathbb{R}^d$. This allows us to support additional user feedback (e.g., ratings) or information (e.g., demographics) as input to $\phi$.

Since Eq. (1) is a policy problem (note breaks affect outcomes), learning requires some form of exploration or experimentation. Here we aim for experimentation to be simple and minimal. Specifically, we assume access to a small number of $N$ additional datasets, $\mathcal{S}^{(j)} = \{(u_k, S_k)\}_{k=1}^{m_j}$, collected for different users, and under composed policies $\pi_j \circ \psi$ for various predetermined $\pi_j$. These datasets can either be given at the onset, or collected as part of the learning process; our only usage of them will be for learning predictors $\hat{y} = f_j(u)$ of engagement rate $y = \frac{1}{T}|S_u|$. Ideally, we would like to make do with only a few $\mathcal{S}^{(j)}$ of small size and that can be gathered concurrently; our results show that even a single additional dataset can be highly informative, and in some cases sufficient.

**Learning to break.** To effectively optimize Eq. (1), learning must anticipate the effects of different breaking policies $\pi \in \Pi$ on future sequences $S_u$ of unobserved users $u$. It will therefore be useful to distinguish between the policy $\pi$ itself, which determines when breaks are applied, and a component for estimating individualized counterfactual engagement rates $\frac{1}{T}|\hat{S}_u(\pi)|$ induced by $\pi$, which will aid in choosing a good policy. Our focus will be on learning simple policies coupled with rich and task-appropriate models of responsive user behavior. In particular, we set $\Pi$ to include all *stationary policies*, $\pi(u) = \pi(p_u)$, which for each user $u$ recommends a break with a time-independent personalized probability $p_u = p(u) \in [0, 1]$. Stationary policies are interpretable, amenable to efficient optimization, and straightforward to implement. As we show, despite their simplicity, such policies can be quite expressive when coupled with an appropriate behavioral model.

## 3 Engagement via 'Predator-Prey' dynamics

To optimize engagement with breaks, we propose to model user behavior as a dynamical system of Lotka-Volterra (LV) Predator-Prey equations. The model operates over continuous dynamics, which has analytic and optimizational benefits; we later make the connection back to discrete sequences.

**Behavioral model.** Our model of user behavior views each user as acting based on two types of time-dependent latent variables: internal *drive* for consumption, denoted by $\lambda(t) \in \mathbb{R}_+$; and intrinsic *interest*, denoted by $q(t) \in [0, 1]$. Intuitively, we expect that engaging experiences will reinforce a user's desire to further consume; conversely, excessive exposure to content should slowly 'erode' her interest—but left alone, will allow it to replenish (Thoman et al., 2011). Thus, drive and interest act as balancing forces. Together, we model the time-dependent relations between $\lambda(t)$ and $q(t)$ as:

$$\frac{d\lambda}{dt} = -\alpha\lambda + \beta\lambda q, \qquad \frac{dq}{dt} = \gamma q(1 - q) - \delta\lambda q \tag{2}$$

where $\theta = (\alpha, \beta, \gamma, \delta) \geq 0$ parameterize the dynamics. For $\lambda$ (drive, or 'predator'), $\alpha$ determines its natural decay rate, and $\beta$ its interest-dependent self-reinforcement rate; for $q$ (interest, or 'prey'), $\gamma$ specifies its natural replenishment rate, and $\delta$ its rate of depletion from consumption. Note the two equations are coupled: $q$ mediates the reinforcement of $\lambda$, and $\lambda$ mediates the depletion of $q$.

---

[1]This is similar in spirit to the 'learning to defer' paradigm (Madras et al., 2018), but in a different context.

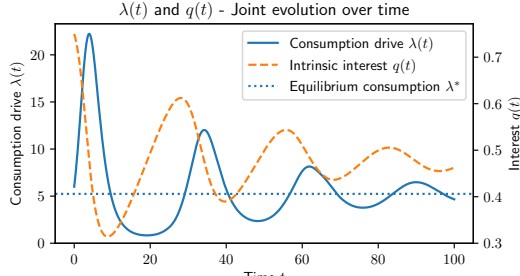
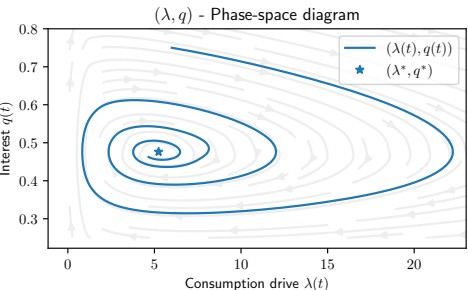

Figure 1: Characteristic LV dynamics (Eq. (2)). **(Left)** Temporal relations between *consumption drive* $\lambda(t)$, *intrinsic interest* $q(t)$, and equilibrium $\lambda^*$. Note how $\lambda$ drops only some time after $q$ has depleted. **(Right)** Evolution of the same system in phase space $(\lambda, q)$, with its equilibrium $(\lambda^*, q^*)$.

**Dynamical properties.** The LV model in Eq. (2) describes consumption as a cycle: when interest $q(t)$ is high, drive to consume $\lambda(t)$ increases, resulting in positive feedback; conversely, when $\lambda(t)$ is high, $q(t)$ decreases, which expresses negative feedback. In general, $\lambda$ grows until interest is too low to sustain consumption, at which point consumption drops sharply, allowing interest to recover—and the cycle repeats. The cycling behavior exhibits oscillations in $\lambda$ and $q$, with one lagging after the other. A typical trajectory is illustrated in Figure 1. Note how the drop in $\lambda$ occurs only some time after $q$ is depleted; hence, anticipating (and preventing) the collapse of $\lambda$ requires knowledge (and conservation) of $q$. Thus, $q$ serves as a resource: necessary for consumption, and of limited supply.

Over time, and if no interventions are applied, the magnitude of oscillations decreases, and the system naturally approaches a stable equilibrium, denoted $(\lambda^*, q^*)$, determined by the system parameters $\theta = (\alpha, \beta, \gamma, \delta)$ and which attracts all initial conditions $\lambda(0), q(0)$ (Takeuchi, 1996). Equilibrium plays a key role in our approach, as it captures the notion of habits, which we aim to improve.

**Engagement maximization as optimal control.** To optimize engagement (Eq. (1)), we propose to optimize $\lambda^*$ as an alternative proxy. Broadly, our approach will be to associate with each user $u$ a dynamical system parameterized by $\theta_u = (\alpha_u, \beta_u, \gamma_u, \delta_u)$, and recommend breaks which lead to high values of the corresponding $\lambda_u^* = \lambda_{\theta_u}^*$. Intuitively, if we think of $\frac{1}{T}|S_u|$ as an 'empirical' rate (determined by the $\Delta t_i$ in $S_u$), then $\lambda_u^*$ is its continuous theoretical counterpart, and so ideally we would like to find a $\theta_u$ for which $\lambda_u^*$ is the limiting behavior of $\frac{1}{T}|S_u|$ (i.e., when $\Delta t \to 0$ and $T \to \infty$). In practice, we expect $\lambda_u^*$ to be a useful target when the observed $\frac{1}{T}|S_u|$ exhibits habits that are 'close enough' to the theoretical equilibrium, and in Sec. 4.3 we make this condition precise.

Since our goal is to optimize a breaking policy, we must also be precise about the way breaks affect dynamics in our model. Towards this, we introduce into Eq. (2) a *control variable*, $p(t) \in [0, 1]$, which acts as a 'gate' that controls the mediation strength between $\lambda(t)$ and $q(t)$ via:

$$\frac{\mathrm{d}\lambda}{\mathrm{d}t} = -\alpha\lambda + \beta\lambda q(1-p), \qquad \frac{\mathrm{d}q}{\mathrm{d}t} = \gamma q(1-q) - \delta\lambda q(1-p) \qquad (3)$$

When $p > 0$, breaking has a dual effect: it decelerates drive $\lambda$, and at the same time, lets $q$ recover.

Our goal will now be to solve the optimal control problem of finding $p(t)$ which maximizes $\lambda^*$. Note that not every dynamic controlling schedule guarantees convergence to some $\lambda^*$. Hence, we focus on *fixed controls*, $p \in [0, 1]$, which we prove converge to equilibrium, and having *closed form*.

**Lemma 1.** *Let* $\theta = (\alpha, \beta, \gamma, \delta)$ *define an LV system as in Eq. (3). Then for any* $p \in [0, 1]$*, we have:*

$$\lambda_\theta^*(p) = \frac{\gamma}{\delta}\frac{1}{1-p}\left(1 - \frac{\alpha}{\beta}\frac{1}{1-p}\right) \quad \text{if } p \in [0, 1-\alpha/\beta], \text{ and zero otherwise.} \qquad (4)$$

Proof in Appendix A.1. For optimization, Lemma 1 is useful since it depicts $\lambda^*$ as a function of $p$, parameterized by $\theta$. Thus, given some $\theta_u$ for user $u$, our objective is to solve the control problem:

$$p_u = \arg\max_{p \in [0,1]} \lambda_{\theta_u}^*(p) \qquad (5)$$

Optimizing engagement now reduces to solving Eq. (5); given $p_u$, we make the connection back to discrete time by setting the breaking policy to be $\pi(p_u)$, which recommends breaks at rate $p_u$.

# 4 Learning optimal breaking policies

We now turn to presenting our learning algorithm. Our approach to optimizing engagement consists of two steps: (i) associating with each user $u$ a set of LV parameters $\theta_u$, and then (ii) finding $p_u$ which maximizes $\lambda_{\theta_u}^*(p)$, and plugging into $\pi(p_u)$. In practice, we add an intermediate prediction step, which allows us to 'shortcut' directly from observations to optimal policies. We conclude with analysis showing when learned policies $\pi(p_u)$ lead to good expected engagement $\mathbb{E}_{u \sim D} \mathbb{E}_{\mathrm{TPP}}\left[\frac{1}{T}|S_u|\right]$.

## 4.1 Embedding users in LV space

To find $\theta_u$, a seemingly reasonable approach would be to fit an LV trajectory to the initial sequence $S_u^0$, i.e., by solving $\min_\theta \sum_i |\Delta t_i - \lambda_\theta(t_i)|$ for the observed $t_i \in S_u^0$. This can be interpreted as embedding $S_u^0$ in 'LV-space' by finding the nearest continuous trajectory, for which $\theta$ provides a compact representation. However, a key issue with this approach is that $S_u^0$ contains *past* observations made under a *single* policy $\pi(p)$ (e.g., the 'clean' policy $\pi(0)$). To optimize *future* engagement, the learned $\theta_u$ must correctly account for the affects of *general* $p$ on possbile ensuing sequences $S_u$.

As an alternative, we propose to find $\theta_u$ by fitting the entire *equilibrium curve* of $\lambda_\theta^*(p)$ (Eq. (4)). Formally, for each $p$, define the *expected empirical engagement rate* $\bar{\lambda}_u(p)$ as:

$$\bar{\lambda}_u(p) = \mathbb{E}_\pi\left[\tfrac{1}{T}|S_u|\right], \qquad S_u \sim \mathrm{TPP}_{\pi(p) \circ \psi}(u; T) \tag{6}$$

i.e., $\bar{\lambda}$ is the rate of *counterfactual* future trajectories for all possible choices of $p$. Ideally, we would like to find $\theta$ for which the learned curve $\lambda_\theta^*(p)$ closely aligns with that of $\bar{\lambda}_u(p)$ across $p \in [0, 1]$:

$$\bar{\theta}_u = \arg\min_\theta \|\bar{\lambda}_u - \lambda_\theta^*\| \tag{7}$$

for some function norm $\|\cdot\|$, and for which $\lambda_{\bar{\theta}_u}^*$ and $\bar{\lambda}_u$ have similar maximizing $p$ (since we aim for optimizing $\lambda_\theta^*$ to work well for $\bar{\lambda}_u$). Unfortunately, $\bar{\lambda}_u$ is a theoretical construct, and so $\bar{\theta}_u$ cannot be obtained from observations. Hence, we propose to replace $\bar{\lambda}_u$ with a finite set of *predictions*.

**The role of prediction.** Recall our input consists of a primary dataset $\mathcal{S}^{(0)} = \mathcal{S} = \{(u_i, S_i)\}_{i=1}^m$, as well as additional 'experimental' datasets $\mathcal{S}^{(j)}$ collected under different $\pi_j = \pi(p_j)$ and of sizes $m_j$ for $j = 1, \ldots, N$. We can use these to learn individualized, policy-specific predictors $f_j(u) = f_{p_j}(u)$, trained to predict for each user $u$ her engagement rate $y = \frac{1}{T}|S_u|$ under $\pi_j$. For example, if we train $f_j$ to minimize the squared error $\sum_k (f_j(u_k) - y_k)^2$ on pairs $(u_k, y_k) \in \mathcal{S}^{(j)}$, then $f_j(u)$ should be a reasonable estimator of the expected $\bar{\lambda}_u(p_j)$. Hence, for a given $u$, a finite set of pairs $\{(p_j, f_j(u))\}_{j=1}^N$ gives points to which we can fit $\theta$ to $\lambda_\theta^*$. Our final criterion for choosing $\hat{\theta}_u$ is:

$$\hat{\theta}_u = \arg\min_\theta \|\boldsymbol{f}(u) - \lambda_\theta^*\| = \arg\min_\theta \sum_{j=1}^N \left(f_j(u) - \lambda_\theta^*(p_j)\right)^2 \tag{8}$$

given here with the $\ell_2$ vector norm, and where $\boldsymbol{f}(u) = (f_1(u), f_2(u), \ldots, f_N(u)) \in \mathbb{R}_+^N$. As we will see next, optimizing over $p$ requires only the ratios $\alpha/\beta$ and $\gamma/\delta$, in which $\lambda^*$ is quadratic. Hence, Eq. (8) can be efficiently solved using a polynomial Non-Negative Least Squares (NNLS) regression solver (Chen and Plemmons, 2010).

**The role of experimentation.** Two parameters control the goodness of fit for $\hat{\theta}_u$: the number of experimental datasets, $N$, and their sizes, $m_j$ for $j \in [N]$. In general, increasing $N$ provides more data points for solving Eq. (8), and increasing each $m_j$ reduces noise for that point (i.e., $f_u(p)$ should be closer to $\bar{\lambda}_u$). But experimentation is costly, and so in reality $N$ and the $m_j$ may be small. As motivation, we next show that under realizability and for accurate predictions, $N = 2$ suffices.

**Proposition 1.** *Fix $N = 2$, and let $p_0, p_1 \in [0, 1 - \alpha/\beta]$. For a user $u$, if (i) exists $\theta_u$ s.t. $\bar{\lambda}_u = \lambda_{\theta_u}^*$, and (ii) $f_i(u) = \bar{\lambda}_u(p_i)$ for $i = 1, 2$, then $\hat{\theta}_u$ is optimal, i.e., solving Eq. (8) gives $\hat{\theta}_u = \bar{\theta}_u$.*

Proof in Appendix A.1, and relies on Lemma 1. Next, we discuss how to obtain $\pi(u)$ from $\theta_u$.

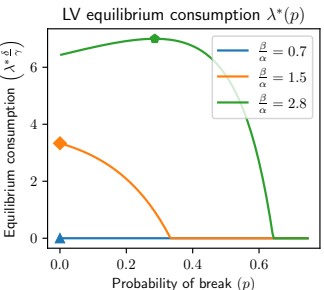 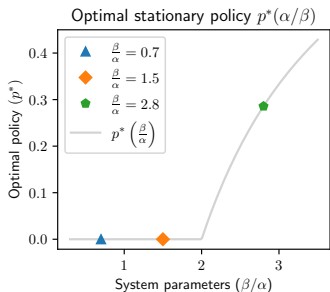 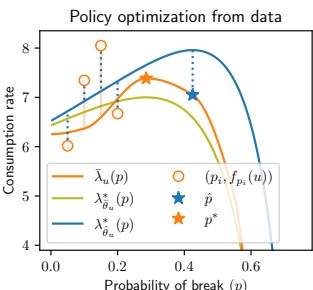

Figure 2: **(Left)** Equilibrium curves $\lambda^*(p)$ and optimal policies $p^*$ (*markers*) for user types ($\theta_u$) that: benefit from breaks (*green*), do not require breaks (*orange*), and will inevitably churn (*blue*). **(Center)** The optimal policy $p^*$ for all $\beta/\alpha$, showing a second-order phase change at $\beta/\alpha = 2$. **(Right)** An illustration of the true counterfactual engagement curve (*orange*),

## 4.2 From predictions to optimal policies

Recall our goal is to learn an *individualized* breaking policy $\pi$ that for each user $u$ applies an appropriate, personalized breaking schedule $\pi(u)$. To be able to generalize to unseen $u$, the conventional approach is to learn a parameterized policy $\pi(u; \eta)$, where $\eta$ is optimized on training data, and applied to new users at test-time. Our approach, by relying on predictions, circumvents the need to learn parameterized policies: once the predictors $\{f_j(u)\}_j$ have been learned, the policy problem decomposes over users, and optimal policies $\pi(u)$ are determined independently for each user.

**Optimal policies.** Given $\theta_u$, our next result establishes a closed form solution for the optimal $p_u$. Note that Eq. (4) shows $\lambda_\theta^*(p)$ is piece-wise polynomial in $z = 1/(1-p)$. Solving for $z$, we get:

**Lemma 2.** *Let $\theta = (\alpha, \beta, \gamma, \delta)$ define an LV system as in Eq. (3). Then the optimal $p^*$ is given by:*

$$p^*(\theta) = \arg\max_{p \in [0,1]} \lambda_\theta^*(p) = 1 - 2\frac{\alpha}{\beta} \quad \text{if} \quad \frac{\alpha}{\beta} \leq \frac{1}{2}, \text{ and zero otherwise.} \tag{9}$$

Proof in Appendix A.1. Hence, once $\theta_u$ is found, our learned policy is defined as:

$$\pi(u) = \pi(\hat{p}_u), \quad \text{where} \quad \hat{p}_u = p^*(\hat{\theta}_u) \tag{10}$$

Altogether, to compute $\hat{p}_u$, the least squares approach in Eq. (8) is applied to obtain the polynomial coefficients $\left(\frac{\gamma_u}{\delta_u}, \frac{\alpha_u \gamma_u}{\beta_u \delta_u}\right)$, then $\frac{\alpha_u}{\beta_u}$ is estimated from their ratio, and then plugged into Eq. (9) to obtain $\hat{p}_u$. For $N = 2$, $\pi(u)$ has a closed-form formulation as a function of predictions (Appendix A.2).

Additionally, note that by Eq. (10), $\hat{p}_u$ exhibits a phase transition at $\frac{\alpha_u}{\beta_u} = \frac{1}{2}$, below which $\hat{p}_u = 0$.

**Corollary 1.** *In LV space, $\pi(u)$ partitions users to those who require breaks, and those who don't.*

**Corollary 2.** *In the realizable case of Prop. 1, $\pi(\hat{p}_u)$ idempotently improves over the myopic $\pi(0)$.*

Thus, the optimal policy can be interpreted as suggesting breaks only when it deems them necessary. Figure 2 illustrates $\lambda^*(p)$ curves, phase shift, and optimal and learned policies for various user types.

## 4.3 Theoretical guarantees

We are now ready to state our main theoretical result, which bounds the expected long-term engagement obtained by our learned policy, $\hat{\pi}$. Our bound shows that the gap between $\hat{\pi}$ and the optimal policy, $\pi^{\mathrm{opt}}$, is goverened by three additive terms, each relating to a different aspect of our approach: modeling error ($\varepsilon_{\mathrm{LV}}$), predictive error ($\varepsilon_{\mathrm{pred}}$), and deviation from expected behavior ($\varepsilon_{\mathrm{dev}}$). A description and interpretation of each term follows shortly. For simplicity, we focus on $N = 2$.

**Theorem 1** (Informal). *Let $p_0, p_1 \in [0, 1]$, and denote by $\pi^{\mathrm{opt}} \in \Pi$ be the optimal stationary policy. Then for the learned breaking policy $\hat{\pi}$, we have:*

$$\mathbb{E}_{u, \pi^{\mathrm{opt}}}\left[\frac{1}{T}|S_u|\right] - \mathbb{E}_{u, \hat{\pi}}\left[\frac{1}{T}|S_u|\right] \leq \frac{\eta_{\mathrm{TPP}}}{|p_1 - p_0|}(\varepsilon_{\mathrm{LV}} + \varepsilon_{\mathrm{pred}} + \varepsilon_{\mathrm{dev}})$$

*where $\eta_{\mathrm{TPP}}$ is a TPP-specific constant scale factor.*

Formal statement, precise definitions, and proof are given in Appendix A.3. The proof consists of three main steps: We begin with a clean LV system at $T = \infty$, and quantify the downstream effect of perturbing the optimal policy. Then, we plug in the learned policy, and bound the gap due to predictive errors and finite $T$. The final step makes the transition from continuous dynamics to the discrete TPP. We next detail the role of each of the five terms in the bound, and how they can be controlled.

- **Predictive error**: Since targets $y = \frac{1}{T}|S_u|$ are real, $\varepsilon_{\mathrm{pred}}$ is simply the expected regression error over users, measured here in RMSE. As is standard, $\varepsilon_{\mathrm{pred}}$ can be reduced by increasing the number of samples $m$, or by learning more expressive predictive functions $f$ (e.g., larger neural nets).

- **Modeling error**: LV dynamics permit tractable learning; but as any hypothesis class, this trades off with model capacity. Here, $\varepsilon_{\mathrm{LV}}$ quantifies the error due limited expressive power. Further reducing $\varepsilon_{\mathrm{LV}}$ can be achieved by consider richer dynamic models—a challenge left for future work.

- **Deviation from expectation**: The learned $\hat{p}_u$ rely on predicted equilibrium, but are trained on finite-horizon data. In expectation, $\varepsilon_{\mathrm{dev}}$ captures how finite sequences deviate from their mean. As a rule of thumb, we expect larger $T$ to reduce this form of noise, but this cannot be guaranteed.

- **Sensitivity** : For $N = 2$, the term $|p_1 - p_0|$ quantifies the added value of exploring beyond the default breaking policy of $p_0$. Intuitively, when the points are farther away, fitting the equilibrium curve is easier. Thus, for the likely case of $p_0 = 0$, the experimental breaking rate $p_1$ should be chosen to balance between performance gain and overexposure of experimental subjects to breaks.

## 5 Experiments

We conclude with an empirical evaluation of our approach on semi-synthetic data. We use the MovieLens 1M dataset to generate recommendations and simulate user behavior, this enabling us to evaluate counterfactual outcomes under different policies. See Appendix B for additional details.

### 5.1 Experimental setup

**Data.** The MovieLens 1M dataset (Harper and Konstan, 2015) includes 1,000,209 ratings provided by 6,040 users and for 3,706 items, which we use to obtain features, determine the dynamics, and emulate $\phi$. We sample and hold out 30% of all ratings $r_{ux}$ via user-stratified sampling, to which we apply Collaborative Filtering (CF) to get user features $u$ and item features $x$ that approximate $u^\top x \approx r_{ux}$ ($d = 8$, RMSE $= 0.917$, $r \in [1, 5]$). This mimics a process where $\phi$ is based on items recommended by $\psi$ and rated by users. We then take the remaining data points and randomly assign 1,000 users to the test set, on which we evaluate policies. The remaining users are assigned to the train set, which is then further partitioned into the main $\mathcal{S}$ and the different $\mathcal{S}^{(j)}$ per experimental condition.

**Recommendation policy and user behavior.** Following Kalimeris et al. (2021); Hron et al. (2022), we set $\psi$ to recommend based on learned softmax scores, $\mathrm{softmax}(u^\top x)$, taken over all non held-out $x$ having ratings for $u$. For user behavior, we generate discrete interaction sequences $S_u$ in a way that relates to our dynamic model, but is nonetheless quite distinct. Specifically, each user $u$ is associated with *discrete-time* latent-state variables, updated via an LV discretization scheme (see Appendix B.5). These, together with recommended items $x_i = x(t_i)$, determine consecutive $\Delta t_i \in S_u$. Importantly, we let $\Delta t_i$ depend on $u$'s rating for $x_i$, namely $r_{u,x_i}$, which we interpret as $u$'s utility from consuming $x_i$. We parameterize this dependence via 'immediate', item-specific consumption gain parameters $\beta_{ui}$ that depend on $r_{u,x_i}$, and for simplicity set $\alpha_u, \gamma_u, \delta_u$ to be fixed.

Note that since $\beta_{ui}$ depends on the recommended $x_i$, it varies in time, and hence there is no single $\beta$ that underlies the dynamics: even in the limit ($\Delta t \to 0, T \to \infty$), user behavior cannot be described by a continuous LV system, which implies $\varepsilon_{\mathrm{LV}} > 0$ (see Appendix Figure 5). Since the baseline RMSE is high, we set $\beta_{ui} \propto \tilde{r}_{u,x_i}^2$, where $\tilde{r}_{u,x_i} = \kappa r_{u,x_i} + (1 - \kappa)u^\top x$, so that $\kappa$ interpolates between predicted ratings ($\kappa = 0$) and true ratings ($\kappa = 1$). This allows us to (indirectly) control $\varepsilon_{\mathrm{pred}}$. For all experiments we use $T = 100$, and so expect a roughly fixed $\varepsilon_{\mathrm{dev}} > 0$.

**Methods.** We compare our approach (LV) to several baselines: (i) a `default` policy which myopically optimizes for immediate engagement (and so does not break); (ii) a 'safety switch' policy

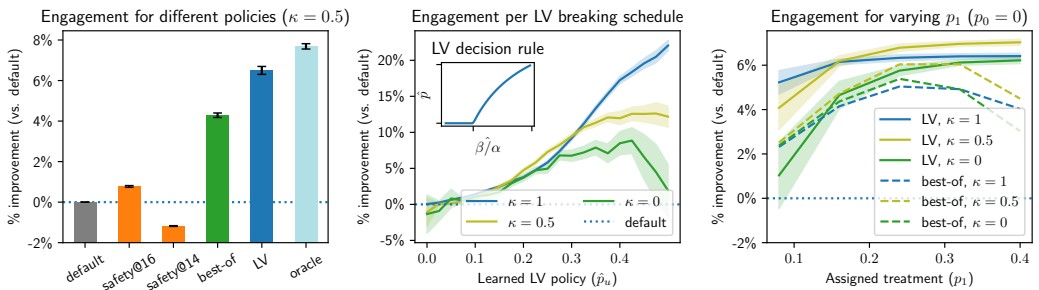

Figure 3: Results on the MovieLens 1M dataset. **(Left)** Performance gain of different approaches (relative to `default` policy). **(Center)** Performance of `LV` by user group, partitioned by learned policies $\hat{p}_u$. **(Right)** Sensitivity to an increasingly aggressive experimental $p_1$ ($N = 2, p_0 = 0$).

(`safety@`$\tau$) that breaks once consumption surpasses a threshold $\tau$; (iii) a prediction-based policy (`best-of`) that chooses the best observed $p_u = \arg\max_{p_j} f_j(u)$ (rather than optimizing over $p_u \in [0,1]$); and (iv) an `oracle` benchmark which directly optimizes the (generally unknown) true underlying dynamics. We measure mean long-term engagement rate (LTE) for each approach, and report averages and standard errors computed over 10 random splits. Performance is measured relative to the `default` baseline as it represents no change in policy (typical absolute values are LTE $\approx 10$).

## 5.2 Results and analysis

**Main results.** Figure 3 (left) compares the performance of our method to other policies. Here we set $p_0 = 0$, use $N = 3$ with $p_j \in \{0.05, 0.1, 0.15\}$, and consider $\kappa = 0.5$ (note $\kappa$ affects all policies). As can be seen, our approach significantly improves over `default` (+6.5%). For `safety@`$\tau$, improvement over the optimal $\tau = 16$ (+5.73%; chosen in hindsight) shows the importance of being preemptive; for the slightly smaller $\tau = 14$, breaks are harmful. The gap from `best-of` (+2.21%) quantifies the gain from the optimization step in Eq. (9), and the close performance to `oracle` (-1.19%) suggests that optimizing the empirical curve (Eq. (8)) works well as as proxy.

**User types.** Figure 3 (center) shows for our approach how gain in LTE varies across learned breaking policies $\hat{p}_u > 0$. For increasingly-accurate predictions ($\kappa \in \{0, 0.5, 1\}$), the main plot shows performance gains for each group of users, partitioned by their $\hat{p}_u$ values (binned; plot shows average and unit standard deviation per bin). Gains until $\hat{p}_u \leq 0.15$ are mild, but for $\hat{p}_u > 0.15$, the general trend is positive: users who are deemed to require more frequent breaks, benefit more from breaking. Gains until $\hat{p}_u \leq 0.3$ increase for all $\kappa$, but for $\hat{p}_u > 0.3$, extrapolation becomes difficult: note the higher variation within each $\kappa$, as well as significant differences across $\kappa$. This highlights the importance of accurate predictions for inferring optimal $\hat{p}_u$ when the experimental $p_i$ are small. The inlaied plot shows that, in line with our theory, $\hat{p}_u$ exhibits an empirical phase shift in the estimated $\hat{\theta}_u$.

**Treatments.** Figure 3 (right) shows the effects of experimental treatments on performance. Focusing on $N = 2$, we fix $p_0 = 0$, and consider increasingly aggressive experimentation by varying $p_1 \in (0, 0.4]$. For our approach, increasing $p_1$ helps, which is anticipated by our theoretical bound. For the `best-of` approach, larger $p_1$ also helps, but exhibits population-level optimum ($p_1 \approx 0.24$), which is easy to 'overshoot'. Note that when prediction accuracy is low ($\kappa = 0$), experimentation is essential: if $p_1$ is not sufficiently large, then performance can sharply deteriorate.

## 6 Discussion

Our paper studies the novel problem of learning optimal breaking policies for recommendation. We posit a tight connection between long-term engagement and user well-being, and argue that optimizing the former requires careful consideration of the latter. Viewing cognitive capacity as a limited but conservable resource, we propose Lotka-Volterra dynamics as a behavioral model that enables effective, responsible, and sustainable optimization of recommendation ecosystems. As any

policy task that involves humans, care must be taken regarding potential risks. While our experiments show that optimizing engagement also improves well-being, this need not always be the case; in fact, merely measuring well-being in reality is challenging. Breaks, as interventions, are presumably 'safe', in the sense that at worst they may lead to suboptimal performance (for system) or satisfaction (for users). But breaks can also be used nefariously, e.g., by enabling temporally-varying rewards (Eyal, 2019). As such, breaking policies should be deployed accountably and transparently.

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

# A Deferred proofs

## A.1 Properties of Lotka-Volterra systems

**Definition A.1** (Static policy equilibrium). *Let $\lambda(t), q(t)$ denote a Lokta-Volterra model characterized by parameters $\theta = (\alpha, \beta, \gamma, \delta) \in \mathbb{R}^4_+$, as defined in Equation 2. Let $p \in [0, 1]$, and denote by $\pi_p$ the static policy corresponding to $p$. For $\lambda(0), q(0) > 0$, the static equilibrium of the system is defined as:*

$$\lambda^*(p; \theta) = \lim_{t \to \infty} \lambda(t)$$
$$q^*(p; \theta) = \lim_{t \to \infty} q(t)$$

We denote $\lambda^*(p) = \lambda^*(p; \theta)$ when $\theta$ is clear from the context. We denote $\lambda^*(p; u) = \lambda^*(p; \theta_u)$ when a user $u \in \mathcal{U}$ characterized by parameters $\theta_u$ is given and clear from the context.

**Proposition A.1** (Global stability). *$\lambda^*(p; \theta)$ exists and uniquely defined for all $\theta \in \mathbb{R}^4_+$, $p \in [0, 1]$ and for all initial conditions $\lambda(0), q(0) > 0$.*

*Proof.* See (Takeuchi, 1996, Section 3.2). $\square$

**Lemma A.1** (Equilibrium of LV behavioral model. Formal proof of Lemma 1). *Assume a Lokta-Volterra model characterized by $\theta = (\alpha, \beta, \gamma, \delta) \in \mathbb{R}^4_+$, and let $p \in [0, 1]$ denote the proportion of interactions in which a forced break is served. The static equilibrium of the model under static policy $\pi_p$ is given by:*

$$\lambda^*(p) = \begin{cases} \frac{\gamma}{\delta} \frac{1}{1-p} \left(1 - \frac{\alpha}{\beta} \frac{1}{1-p}\right) & p \in \left[0, 1 - \frac{\alpha}{\beta}\right] \\ 0 & \text{otherwise} \end{cases}$$

$$q^*(p) = \begin{cases} \frac{\alpha}{\beta} \frac{1}{1-p} & p \in \left[0, 1 - \frac{\alpha}{\beta}\right] \\ 1 & \text{otherwise} \end{cases}$$

*Proof.* The LV dynamical system is given by Equation 3:

$$\frac{\mathrm{d}\lambda}{\mathrm{d}t} = -\alpha\lambda + \beta(1-p)\lambda q$$
$$\frac{\mathrm{d}q}{\mathrm{d}t} = \gamma q(1-q) - \delta(1-p)\lambda q$$

when $p \in \left[0, 1 - \frac{\alpha}{\beta}\right]$ we equate $\frac{\mathrm{d}\lambda}{\mathrm{d}t} = 0$, $\frac{\mathrm{d}q}{\mathrm{d}t} = 0$ and obtain the result. The solution is guaranteed to be valid, as both $\lambda^*(p) > 0$ and $q^*(p) \in [0, 1]$.

Conversely, when $p \notin \left[0, 1 - \frac{\alpha}{\beta}\right]$, there exists $\epsilon > 0$ such that $\frac{\mathrm{d}}{\mathrm{d}t} \log \lambda < -\epsilon < 0$ for all $\lambda > 0$, $q \in [0, 1]$. From this we obtain that $\log \lambda(t)$ tends towards $-\infty$, and therefore $\lambda(t)$ tends towards 0, and $\lambda^*(p) = 0$ as required. When $\lambda(t)$ is close to zero, the interaction terms vanish in the $\frac{\mathrm{d}q}{\mathrm{d}t}$ equation, and $q(t)$ grows logistically towards 1. $\square$

**Proposition A.2** (Equilibrium bounds). *For a Lotka-Volterra model, the static equilibrium $\lambda^*(p)$ is bounded by:*

$$0 \leq \lambda^*(p) \leq \frac{\beta\gamma}{4\alpha\delta}$$

*Proof.* Denote $x = \frac{1}{1-p}$. From Lemma A.1, for $x \in \left[1, \frac{\beta}{\alpha}\right]$ the equilibrium consumption $\lambda^*(x)$ is given by:

$$\lambda^*(x) = \frac{\gamma}{\delta} x \left(1 - \frac{\alpha}{\beta} x\right)$$

and is zero otherwise. The equilibrium is a quadratic function of $x$ with roots $x \in \left\{0, \frac{\beta}{\alpha}\right\}$, and therefore attains its maximum at $x = \frac{\beta}{2\alpha}$. Plugging back the maximizing $x$ into $\lambda^*$ we obtain

the upper bound. Lower bound is attained as the equilibrium in Lemma A.1 is clipped by 0 from below. □

**Lemma A.2** (Optimal static policy. Formal proof of Lemma 2). *The optimal static policy for a Lotka-Volterra system is given by:*

$$p_{\mathrm{opt}} = \begin{cases} 1 - 2\frac{\alpha}{\beta} & \frac{\alpha}{\beta} \leq \frac{1}{2} \\ 0 & \frac{\alpha}{\beta} > \frac{1}{2} \end{cases}$$

*And the optimal equilibrium engagement rate is given by:*

$$\lambda_{\mathrm{opt}}^* = \begin{cases} \frac{\beta\gamma}{4\alpha\delta} & \frac{\alpha}{\beta} \leq \frac{1}{2} \\ \frac{\gamma}{\delta}\left(1 - \frac{\alpha}{\beta}\right) & \frac{\alpha}{\beta} > \frac{1}{2} \end{cases}$$

*Proof.* Denote $x = \frac{1}{1-p}$. From Proposition A.2, the global maximum of $\lambda^*(x)$ is attained at $x = \frac{\beta}{2\alpha}$. Consider two cases: When $\frac{\alpha}{\beta} \leq \frac{1}{2}$, we obtain that $x_{\mathrm{opt}} = \frac{\beta}{2\alpha} \geq 1$, and therefore $p_{\mathrm{opt}} = 1 - \frac{1}{x} \in [0, 1]$. From this we obtain that in this case the global maximum is attained on the simplex, and given by the formula from Proposition A.2. Conversely, when $\frac{\alpha}{\beta} > \frac{1}{2}$, we obtain $p = 1 - \frac{1}{x} < 0$, and therefore $x_{\mathrm{opt}}$ translates to a negative value of $p$. As $\lambda^*(p)$ is uni-modal, the optimal policy restricted to the simplex $[0, 1]$ in this case is attained on the closest boundary point $p = 0$.

Figure 2 provides graphical intuition for this proof (left and center subplots). □

**Proposition A.3** (Inference of $\alpha/\beta$ from two-treatment equilibrium data. Formal proof of Proposition 1). *Let $\lambda(t), q(t)$ be a Lokta-Volterra model, let $p_1, p_2 \in [0, 1]$. Denote by $\lambda^*(p_1), \lambda^*(p_2)$ the static equilibrium rates corresponding to static policies $\pi_{p_1}, \pi_{p_2}$, and assume $\lambda^*(p_1), \lambda^*(p_2) > 0$. The parameter ratio $\frac{\alpha}{\beta}$ is given by the following formula:*

$$\frac{\alpha}{\beta} = \frac{(1 - p_2)\lambda^*(p_2) - (1 - p_1)\lambda^*(p_1)}{\frac{1}{1-p_1} - \frac{1}{1-p_2}}$$

*Proof.* From Lemma A.1, the equilibrium consumption $\lambda^*(p)$ is given by:

$$\lambda^*(p) = \frac{\gamma}{\delta}\frac{1}{1-p}\left(1 - \frac{\alpha}{\beta}\frac{1}{1-p}\right)$$

$$= \frac{\gamma}{\delta}\frac{1}{1-p} - \frac{\alpha}{\beta}\frac{\gamma}{\delta}\left(\frac{1}{1-p}\right)^2$$

When $\lambda^*(p_i)$ is observed for different policies $p_1, \ldots, p_m \in \left[0, 1 - \frac{\alpha}{\beta}\right]$, we obtain a polynomial regression problem for the parameters $\frac{\alpha}{\beta}$ and $\frac{\alpha}{\beta}\frac{\gamma}{\delta}$, which can be solved e.g using Non-Negative Least Squares.

When $m = 2$, we obtain a system of two linear equations. Apply Cramer's rule to obtain:

$$\frac{\gamma}{\delta} = \frac{\frac{\lambda^*(p_2)}{(1-p_1)^2} - \frac{\lambda^*(p_1)}{(1-p_2)^2}}{\frac{1}{(1-p_1)(1-p_2)^2} - \frac{1}{(1-p_1)^2(1-p_2)}} = \frac{(1 - p_2)^2\lambda^*(p_2) - (1 - p_1)^2\lambda^*(p_1)}{p_2 - p_1} \tag{11}$$

$$\frac{\alpha}{\beta}\frac{\gamma}{\delta} = \frac{\frac{\lambda^*(p_2)}{(1-p_1)} - \frac{\lambda^*(p_1)}{(1-p_2)}}{\frac{1}{(1-p_1)(1-p_2)^2} - \frac{1}{(1-p_1)^2(1-p_2)}} = (1 - p_1)(1 - p_2)\frac{(1 - p_2)\lambda^*(p_2) - (1 - p_1)\lambda^*(p_1)}{p_2 - p_1} \tag{12}$$

And therefore $\frac{\alpha}{\beta}$ is given by:

$$\frac{\alpha}{\beta} = \frac{\frac{\lambda^*(p_2)}{(1-p_1)} - \frac{\lambda^*(p_1)}{(1-p_2)}}{\frac{\lambda^*(p_2)}{(1-p_1)^2} - \frac{\lambda^*(p_1)}{(1-p_2)^2}} = (1 - p_1)(1 - p_2)\frac{(1 - p_2)\lambda^*(p_2) - (1 - p_1)\lambda^*(p_1)}{(1 - p_2)^2\lambda^*(p_2) - (1 - p_1)^2\lambda^*(p_1)}$$

□

## A.2 Model fitting from engagement predictions

**Notations.** In this section only, we use the common notation $q = 1 - p$ to denote complementary probabilities.

**Definition A.2** (Empirical value of $\alpha/\beta$). *For single-channel experiments with forced-break probabilities $p_1, p_2$, denote $\lambda_i = \lambda^*(p_i)$, $f_i = f_{p_i}(u)$, $q_i = 1 - p_i$. The empirical value of the $\frac{\alpha}{\beta}$ parameter is given by the following formula:*

$$\frac{\hat{\alpha}}{\beta} = \frac{q_1 q_2 (q_1 f_1 - q_2 f_2)}{q_1^2 f_1 - q_2^2 f_2}$$

**Proposition A.4** ($\alpha/\beta$ estimation error from prediction errors). *Given a single-channel Lokta-Volterra system with parameter $\frac{\alpha}{\beta} \geq 1$. Let $p_1, p_2 \in \left[1, \frac{\alpha}{\beta}\right]$, denote $\lambda_i^* = \lambda^*(p_i) \in \mathbb{R}_+$, and let $f_i = \lambda_i^* + \varepsilon_i$ be the predicted engagement rates corresponding to $p_1, p_2$. When $|\varepsilon_1|, |\varepsilon_2| \leq \varepsilon \leq \frac{\gamma}{\delta} \frac{|p_1 - p_2|}{4}$, the estimation error is bounded by:*

$$\left| \frac{\alpha}{\beta} - \frac{\hat{\alpha}}{\beta} \right| \leq \frac{\varepsilon}{|p_1 - p_2|} \frac{\beta \delta}{\alpha \gamma}$$

*Proof.* denote $q_i = 1 - p_i$. The value of $\frac{\alpha}{\beta}$ is given by Proposition A.3:

$$\frac{\alpha}{\beta} = \frac{q_1 q_2 (q_1 \lambda_1^* - q_2 \lambda_2^*)}{q_1^2 \lambda_1^* - q_2^2 \lambda_2^*}$$

And the estimator for $\frac{\alpha}{\beta}$ is obtained by replacing the true value with their predictions:

$$\begin{aligned}
\frac{\hat{\alpha}}{\beta} &= \frac{q_1 q_2 (q_1 f_1 - q_2 f_2)}{q_1^2 f_1 - q_2^2 f_2} \\
&= \frac{q_1 q_2 (q_1 (\lambda_1^* + \varepsilon_1) - q_2 (\lambda_2^* + \varepsilon_2))}{q_1^2 (\lambda_1^* + \varepsilon_1) - q_2^2 (\lambda_2^* + \varepsilon_2)}
\end{aligned}$$

The estimation error is given by:

$$\begin{aligned}
\left| \frac{\alpha}{\beta} - \frac{\hat{\alpha}}{\beta} \right| &= \left| \frac{q_1^2 q_2^2 (q_1 - q_2)(\varepsilon_2 \lambda_1^* - \varepsilon_1 \lambda_2^*)}{(q_1^2 \lambda_1^* - q_2^2 \lambda_2^*)(q_1^2 \lambda_1^* - q_2^2 \lambda_2^* - (q_1^2 \varepsilon_1 - q_2^2 \varepsilon_2))} \right| \\
&= \underbrace{(q_1 q_2)^2}_{\equiv (i)} \underbrace{\left| \frac{q_1 - q_2}{q_1^2 \lambda_1^* - q_2^2 \lambda_2^*} \right|}_{\equiv (ii)} \underbrace{\left| \varepsilon_2 \lambda_1^* - \varepsilon_1 \lambda_2^* \right|}_{\equiv (iii)} \underbrace{\left| \frac{1}{q_1^2 \lambda_1^* - q_2^2 \lambda_2^* - (q_1^2 \varepsilon_1 - q_2^2 \varepsilon_2)} \right|}_{\equiv (iv)}
\end{aligned}$$

We now proceed to bound each factor:

- For (i), the term $(q_1 q_2)^2$ is bounded by 1 since $q_1, q_2 \in [0, 1]$.

- For (ii), the term $\left| \frac{q_1 - q_2}{q_1^2 \lambda_1^* - q_2^2 \lambda_2^*} \right|$ is equal to $\left( \frac{\gamma}{\delta} \right)^{-1}$ by Eq. (11).

- For (iii), from Proposition A.2 we obtain the bound $0 \leq \lambda_i^* \leq \frac{\beta \gamma}{4 \alpha \delta}$, and therefore the term $|\varepsilon_2 \lambda_1^* - \varepsilon_1 \lambda_2^*|$ is bounded by $2 \left( \frac{\beta \gamma}{4 \alpha \delta} \right) \varepsilon = \frac{\beta \gamma}{2 \alpha \delta} \varepsilon$.

- For (iv), the term $\left|\frac{1}{q_1^2\lambda_1^* - q_2^2\lambda_2^* - (q_1^2\varepsilon_1 - q_2^2\varepsilon_2)}\right|$ is equal to:

$$(\text{iv}) \equiv \left|\frac{1}{q_1^2\lambda_1^* - q_2^2\lambda_2^* - (q_1^2\varepsilon_1 - q_2^2\varepsilon_2)}\right|$$

$$= \frac{1}{|p_1 - p_2|}\left|\frac{q_1^2\lambda_1^* - q_2^2\lambda_2^* - (q_1^2\varepsilon_1 - q_2^2\varepsilon_2)}{p_1 - p_2}\right|^{-1}$$

$$= \frac{1}{|p_1 - p_2|}\left|\underbrace{\frac{q_1^2\lambda_1^* - q_2^2\lambda_2^*}{p_1 - p_2}}_{\text{Eq. (11)}} - \frac{q_1^2\varepsilon_1 - q_2^2\varepsilon_2}{p_1 - p_2}\right|^{-1}$$

$$= \frac{1}{|p_1 - p_2|}\left|\frac{\gamma}{\delta} - \frac{q_1^2\varepsilon_1 - q_2^2\varepsilon_2}{p_1 - p_2}\right|^{-1}$$

Note that $\left|\frac{q_1^2\varepsilon_1 - q_2^2\varepsilon_2}{p_1 - p_2}\right| \le \frac{2\varepsilon}{|p_1 - p_2|}$. When $\varepsilon$ is small enough, and specifically when the bound $\varepsilon \le \frac{\gamma}{\delta}\frac{|p_1 - p_2|}{4}$ holds, we obtain:

$$\left|\frac{\gamma}{\delta} - \frac{q_1^2\varepsilon_1 - q_2^2\varepsilon_2}{p_1 - p_2}\right|^{-1} \le \frac{\delta}{\gamma}\left|1 - \frac{1}{2}\right|^{-1} \le 2\frac{\delta}{\gamma}$$

and therefore:

$$(\text{iv}) \le \frac{2}{|p_1 - p_2|}\frac{\delta}{\gamma}$$

Aggregating results (i)-(iv) above, we obtain the overall bound:

$$\left|\frac{\alpha}{\beta} - \frac{\hat{\alpha}}{\beta}\right| = \underbrace{(q_1 q_2)^2}_{\le 1}\underbrace{\left|\frac{q_1 - q_2}{q_1^2\lambda_1^* - q_2^2\lambda_2^*}\right|}_{= \frac{\delta}{\gamma}}\underbrace{\left|\varepsilon_2\lambda_1^* - \varepsilon_1\lambda_2^*\right|}_{\le \frac{\beta\gamma}{2\alpha\delta}\varepsilon}\underbrace{\left|\frac{1}{q_1^2\lambda_1^* - q_2^2\lambda_2^* - (q_1^2\varepsilon_1 - q_2^2\varepsilon_2)}\right|}_{\le \frac{2}{|p_1 - p_2|}\frac{\delta}{\gamma}}$$

$$\le \frac{\varepsilon}{|p_1 - p_2|}\frac{\beta\delta}{\alpha\gamma}$$

$\square$

**Proposition A.5** (Cost of $\alpha/\beta$ estimation error). *Let $\frac{\alpha}{\beta}$ be the engagement ratio parameter of a one-channel Lotka-Volterra system, and let $\left(\frac{\hat{\alpha}}{\beta}\right)$ be an estimate of these parameters. Let $\lambda_{\text{opt}}^*$ be the engagement rate of the optimal static policy, and denote $\lambda^*(x) = \lambda^*(\hat{p}(x))$. When $\left|\frac{\alpha}{\beta} - \left(\frac{\hat{\alpha}}{\beta}\right)\right| \le \min\left\{\frac{\alpha}{2\beta}, 1\right\}$ The price of estimation error is bounded by:*

$$\lambda_{\text{opt}}^* - \lambda^*\left(\left(\frac{\hat{\alpha}}{\beta}\right)\right) \le \left(\frac{\gamma}{\delta}\right)\min\left\{\left(2\frac{\alpha}{\beta}\right)^{-2}\left|\frac{\alpha}{\beta} - \left(\frac{\hat{\alpha}}{\beta}\right)\right|, \left(4\frac{\alpha}{\beta}\right)^{-1}\right\}$$

*Proof.* Denote $r = \frac{\alpha}{\beta}, x = \left(\frac{\hat{\alpha}}{\beta}\right)$, and assume without loss of generality that $\frac{\gamma}{\delta} = 1$ and $r \le 1$. The optimal equilibrium engagement rate is given by:

$$\lambda_{\text{opt}}^* = \begin{cases} \frac{1}{4r} & r \in \left(0, \frac{1}{2}\right] \\ 1 - r & r \in \left(\frac{1}{2}, 1\right] \end{cases}$$

The chosen policy $\hat{p}(x)$ is given by:

$$\hat{p}(x) = \begin{cases} 1 - 2x & x \in \left[0, \frac{1}{2}\right] \\ 0 & \text{otherwise} \end{cases}$$

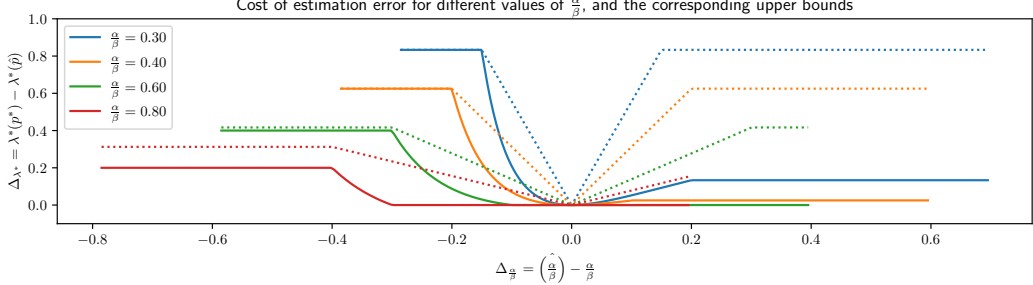

Figure 4: Graphical illustration of Proposition A.5. Cost of estimation error for different values of $\frac{\alpha}{\beta}$, and their corresponding upper bounds given by the claim.

Assume without loss of generality that $x \in \left[0, \frac{1}{2}\right]$, as values of $x$ outside the interval can be clipped to its edges without affecting the result. The equilibrium engagement rate of the selected policy is given by:

$$\lambda^*(x) = \lambda^*\left(\hat{p}(x)\right) = \begin{cases} 0 & x \in \left[0, \frac{r}{2}\right] \\ \frac{1}{2x}\left(1 - \frac{r}{2x}\right) & x \in \left(\frac{r}{2}, \frac{1}{2}\right] \end{cases}$$

Denote $\Delta(x) = \lambda^*_{\text{opt}} - \lambda^*(x)$. We obtain:

$$\Delta(x) = \lambda^*_{\text{opt}} - \lambda^*(x) = \begin{cases} \frac{1}{4r} & r \in \left(0, \frac{1}{2}\right], x \in \left[0, \frac{r}{2}\right] \\ \frac{(x-r)^2}{4x^2 r} & r \in \left(0, \frac{1}{2}\right], x \in \left(\frac{r}{2}, \frac{1}{2}\right] \\ (1-r) & r \in \left(\frac{1}{2}, 1\right], x \in \left[0, \frac{r}{2}\right] \\ (1-r) - \frac{1}{2x}\left(1 - \frac{r}{2x}\right) & r \in \left(\frac{1}{2}, 1\right], x \in \left(\frac{r}{2}, \frac{1}{2}\right] \end{cases}$$

Observe that $\frac{1}{4r} \geq 1 - r$ for all $r \in (0, 1]$, and therefore we obtain for all $x, r$:

$$\Delta(x) \leq \frac{1}{4r} \tag{13}$$

From the convexity of $\Delta(x)$ in the region around $x = r$ we obtain:

$$\Delta(x) \leq \frac{1}{2r^2}|x - r| \tag{14}$$

Finally, combining the two bounds yields the final result. A geometric interpretation of this claim is illustrated in Figure 4. $\qquad\square$

### A.3 Optimal stationary policy from engagement predictions

**Definition A.3** (Expected observable rate). *Let $u \in \mathcal{U}$, $p \in [0, 1]$, and $T > 0$. Let $p \in [0, 1]$, denote the corresponding static policy by $\pi_p$. The expected observable rate $\bar{\lambda}_u(p; T)$ is defined as:*

$$\bar{\lambda}_u(p; u) = \mathbb{E}_\pi\left[\frac{1}{T}\left|\text{TPP}_{\pi_p}(u; T)\right|\right]$$

*where expectation is taken over the stochastic decisions of $\pi_p$.*

**Definition A.4** (Lokta-Volterra approximation of TPP). *Let $u \in \mathcal{U}$, and $T > 0$. Denote by $p^*$ the maximizer of expected observable rate:*

$$p^* = \underset{p \in [0,1]}{\arg\max}\, \bar{\lambda}_u(p; u)$$

*The LV approximation of $\text{TPP}(u; T)$ is defined as:*

$$\theta^*_u = \arg\min_\theta \max_{p \in [0,1]} \left|\bar{\lambda}_u(p; u) - \lambda^*(p; \theta)\right|$$

*such that $\arg\max_p \lambda^*(p; \theta) = p^*$. The corresponding approximation error is defined as:*

$$\varepsilon_{\text{LV}, u} = \max_{p \in [0,1]} \left|\bar{\lambda}_u(p; u) - \lambda^*(p; \theta^*_u)\right|$$

**Notations.** When $u$ is clear from the context, we denote $\theta^* = \theta_u^*$, $\varepsilon_{\mathrm{LV}} = \varepsilon_{\mathrm{LV},u}$. We use $\alpha^*, \beta^*, \ldots$ to refer to the corresponding parts of the Lokta-Volterra parameters vector $\theta^*$.

We are now ready to state and prove the main theorem for this section:

**Theorem A.1** (Regret bound for learned static policy. Formal version of Theorem 1). *Let $p_1, p_2 \in [0,1]$ denote two static forced-break policies, and denote by $\mathcal{U}$ the set of users, and assume they remain engaged under the stationary policies $\pi(p_1)$ and $\pi(p_2)$. Assume $S_u(p;T) \sim \mathrm{TPP}_{\pi_p \circ \psi}(u;T)$, and let $\mu = \left(\max_{u \in \mathcal{U}} \frac{\bar{\gamma}_u}{\delta_u}\right) \cdot \left(\max_{u' \in \mathcal{U}} \frac{\bar{\delta}_{u'}}{\gamma_{u'}}\right)$, $\nu = \max_{u \in \mathcal{U}} \left(\frac{\bar{\beta}_u}{\alpha_u}\right)$.*

*Let $f_{p_1}, f_{p_2} : \mathcal{U} \to \mathbb{R}_+$ be functions predicting $\frac{1}{T}|S_u(p_1;T)|$, $\frac{1}{T}|S_u(p_2;T)|$, respectively. Denote the learned policy by $\hat{p}$, and the optimal policy by $p^*$.*

*If (i) the expected RMSE of $f_{p_1}, f_{p_2}$ is bounded by $\varepsilon_{\mathrm{pred}}$, (ii) the average absolute deviation of $\frac{1}{T}|\mathrm{TPP}(u;T)|$ is bounded by $\varepsilon_{\mathrm{dev}}$, and (iii) the expected LV approximation error of the system is bounded by $\varepsilon_{\mathrm{LV}}$, then the learned policy $\hat{p}$ has bounded regret:*

$$\mathbb{E}_{u,\pi}\left[\left|\tfrac{1}{T}|S_u(p^*;T)| - \tfrac{1}{T}|S_u(\hat{p};T)|\right|\right] \leq \frac{\eta_{\mathrm{TPP}}}{|p_1 - p_2|}(\varepsilon_{\mathrm{pred}} + \varepsilon_{\mathrm{dev}} + \varepsilon_{\mathrm{LV}})$$

*where expectation is taken over stochastic choices of policies, and $\eta_{\mathrm{TPP}} = g(\mu, \nu) \in \mathrm{poly}(\mu, \nu)$.*

*Proof.* By assumption (i), the functions $f_{p_1}, f_{p_2}$ have bounded expected RMSE:

$$\mathbb{E}_u\left[\left(f_{p_i}(u) - \tfrac{1}{T}|S_u(p_i;T)|\right)^2\right] \leq \varepsilon_{\mathrm{pred}}^2 \tag{15}$$

Applying Jensen's inequality with the convex function $\varphi(x) = x^2$ yields:

$$\left(\mathbb{E}_u\left[\left|f_{p_i}(u) - \tfrac{1}{T}|S_u(p_i;T)|\right|\right]\right)^2 \leq \mathbb{E}_u\left[\left(f_{p_i}(u) - \tfrac{1}{T}|S_u(p_i;T)|\right)^2\right]$$

Combining with Eq. (15) and taking the square root, we obtain an upper bound on the expected absolute error:

$$\mathbb{E}_u\left[\left|f_{p_i}(u) - \tfrac{1}{T}|S_u(p_i;T)|\right|\right] \leq \varepsilon_{\mathrm{pred}} \tag{16}$$

Let $\Delta_f = |f_{p_i}(u) - \lambda^*(p_i)|$ apply the triangle inequality to obtain:

$$\begin{aligned}
\Delta_f &= |f_{p_i}(u) - \lambda^*(p_i)| \\
&\leq \left|f_{p_i}(u) - \tfrac{1}{T}|S_u(u;T)|\right| + \left|\tfrac{1}{T}|S_u(u;T)| - \bar{\lambda}(p_i;u)\right| + \left|\bar{\lambda}(p_i;u) - \lambda^*(p_i)\right|
\end{aligned}$$

Denote $\varepsilon_f = \varepsilon_{\mathrm{pred}} + \varepsilon_{\mathrm{dev}} + \varepsilon_{\mathrm{LV}}$. Applying the triangle inequality and using the bounds in Eq. (16) together with assumptions (ii), (iii), we obtain:

$$\begin{aligned}
\mathbb{E}_{u,\pi}[\Delta_f] \leq\, &\mathbb{E}_u\left[\left|f_{p_i}(u) - \tfrac{1}{T}|S_u(u;T)|\right|\right] \\
&+ \mathbb{E}_{u,\pi}\left[\left|\tfrac{1}{T}|S_u(u;T)| - \bar{\lambda}(p_i;u)\right|\right] \\
&+ \mathbb{E}_u\left[\left|\bar{\lambda}(p_i;u) - \lambda^*(p_i;\theta_u^*)\right|\right] \\
\leq\, &\varepsilon_{\mathrm{pred}} + \varepsilon_{\mathrm{dev}} + \varepsilon_{\mathrm{LV}} = \varepsilon_f
\end{aligned} \tag{17}$$

Denote $\theta_u^* = (\alpha, \beta, \gamma, \delta)$. The empirical value $\left(\hat{\tfrac{\alpha}{\beta}}\right)$ of $\left(\tfrac{\alpha}{\beta}\right)$ is given by Definition A.2. Denote the estimation error by $\Delta_{\frac{\alpha}{\beta}} = \left|\left(\hat{\tfrac{\alpha}{\beta}}\right) - \left(\tfrac{\alpha}{\beta}\right)\right|$.

By Proposition A.4, the following pointwise upper bound on $\Delta_{\frac{\alpha}{\beta}}$ applies when $\Delta_f \leq \frac{\gamma}{\delta}\frac{|p_1 - p_2|}{4}$:

$$\Delta_{\frac{\alpha}{\beta}} \leq \frac{\Delta_f}{|p_1 - p_2|}\frac{\beta\delta}{\alpha\gamma} \tag{18}$$

Plugging in the bound on the expected value of $\Delta_f$ into Eq. (18), we obtain in expectation:

$$\begin{aligned}
\mathbb{E}_{u,\pi}\left[\Delta_{\frac{\alpha}{\beta}} \mid \Delta_f \leq \frac{\gamma}{\delta}\frac{|p_1 - p_2|}{4}\right] &\leq \mathbb{E}_{u,\pi}\left[\frac{\Delta_f}{|p_1 - p_2|}\frac{\beta\delta}{\alpha\gamma} \mid \Delta_f \leq \frac{\gamma}{\delta}\frac{|p_1 - p_2|}{4}\right] \\
&\leq \frac{\varepsilon_f}{|p_1 - p_2|}\max_u \frac{\beta\delta}{\alpha\gamma}
\end{aligned} \tag{19}$$

Next, we apply Proposition A.5. Denote $\Delta_{\lambda^*} = \lambda^*(p^*) - \lambda^*(\hat{p})$, and define the following probability event:

$$A = \left(\Delta_f \le \frac{\gamma}{\delta}\frac{|p_1 - p_2|}{4}\right) \text{ and } \left(\Delta_{\frac{\alpha}{\beta}} \le \frac{1}{2\nu}\right)$$

Note that the bound in Proposition A.5 is represented as a minimum between two functions, one linear in $\varepsilon$ and one constant. To leverage this property, apply the law of total expectation:

$$\mathbb{E}_{u,\pi}[\Delta_{\lambda^*}] = \mathbb{E}_{u,\pi}[\Delta_{\lambda^*} \mid A]\mathbb{P}[A] + \mathbb{E}_{u,\pi}[\Delta_{\lambda^*} \mid \bar{A}]\mathbb{P}[\bar{A}] \tag{20}$$

Under $A$, the first term in Eq. (20) can be bounded by the linear term in Proposition A.5. Taking $\mathbb{P}[A] \le 1$ and combining with equation Eq. (18):

$$\mathbb{E}_{u,\pi}[\Delta_{\lambda^*} \mid A]\mathbb{P}[A] \le \mathbb{E}_{u,\pi}[\Delta_{\lambda^*} \mid A]$$

$$\le \mathbb{E}_{u,\pi}\left[\frac{\beta^2\gamma}{2\alpha^2\delta}\Delta_{\frac{\alpha}{\beta}} \mid A\right]$$

$$\le \mathbb{E}_{u,\pi}\left[\frac{\beta^2\gamma}{2\alpha^2\delta}\frac{\Delta_f}{|p_1-p_2|}\frac{\beta\delta}{\alpha\gamma} \mid A\right]$$

$$\le \frac{\nu^3}{2|p_1-p_2|}\varepsilon_f \tag{21}$$

The expectation factor in the second term of Eq. (20) can be bounded by the constant term in Proposition A.5:

$$\mathbb{E}_{u,\pi}[\Delta_{\lambda^*} \mid \bar{A}] \le \frac{1}{4}\max_u \frac{\beta\gamma}{\alpha\delta} \le \frac{\nu}{4}\max_u \frac{\gamma}{\delta} \tag{22}$$

Decompose the probability factor $\mathbb{P}[\bar{A}]$ using the law of total probability:

$$\mathbb{P}[\bar{A}] = \mathbb{P}\left[\Delta_f > \frac{\gamma}{\delta}\frac{|p_1-p_2|}{4}\right] + \mathbb{P}\left[\Delta_{\frac{\alpha}{\beta}} > \frac{1}{2\nu} \mid \Delta_f \le \frac{\gamma}{\delta}\frac{|p_1-p_2|}{4}\right]\mathbb{P}\left[\Delta_f \le \frac{\gamma}{\delta}\frac{|p_1-p_2|}{4}\right]$$

$$\le \mathbb{P}\left[\Delta_f > \frac{\gamma}{\delta}\frac{|p_1-p_2|}{4}\right] + \mathbb{P}\left[\Delta_{\frac{\alpha}{\beta}} > \frac{1}{2\nu} \mid \Delta_f \le \frac{\gamma}{\delta}\frac{|p_1-p_2|}{4}\right]$$

Apply Markov's inequality $\mathbb{P}[|X| \ge a] \le \frac{\mathbb{E}[|X|]}{a}$ on the probabilities to obtain:

$$\mathbb{P}\left[\Delta_f > \frac{\gamma}{\delta}\frac{|p_1-p_2|}{4}\right] \le \mathbb{E}_{u,\pi}[\Delta_f]\left(\frac{\gamma}{\delta}\frac{|p_1-p_2|}{4}\right)^{-1}$$

$$\underset{\text{by Eq. (17)}}{\le} \varepsilon_f\frac{4}{|p_1-p_2|}\max_u \frac{\delta}{\gamma} \tag{23}$$

$$\mathbb{P}\left[\Delta_{\frac{\alpha}{\beta}} > \frac{1}{2\nu} \mid \Delta_f \le \frac{\gamma}{\delta}\frac{|p_1-p_2|}{4}\right] \le \mathbb{E}_{u,\pi}\left[\Delta_{\frac{\alpha}{\beta}} \mid \Delta_f \le \frac{\gamma}{\delta}\frac{|p_1-p_2|}{4}\right]$$

$$\underset{\text{by Eq. (19)}}{\le} \frac{\varepsilon_f}{|p_1-p_2|}\max_u \frac{\beta\delta}{\alpha\gamma}$$

$$\le \frac{\varepsilon_f}{|p_1-p_2|}\nu\max_u \frac{\delta}{\gamma} \tag{24}$$

Plugging back equations Eq. (21), Eq. (22),Eq. (23),Eq. (24) into equation Eq. (20), we obtain bounds for each term:

$$\mathbb{E}_{u,\pi}[\Delta_{\lambda^*}] = \underbrace{\mathbb{E}_{u,\pi}[\Delta_{\lambda^*} \mid A]\mathbb{P}[A]}_{\text{by Eq. (21)}} + \underbrace{\mathbb{E}_{u,\pi}[\Delta_{\lambda^*} \mid \bar{A}]}_{\text{by Eq. (22)}}\underbrace{\mathbb{P}[\bar{A}]}_{\text{by Eq. (23),Eq. (24)}} \tag{25}$$

we obtain:

$$\mathbb{E}_{u,\pi}[\Delta_{\lambda^*}] \le \frac{\varepsilon_f}{|p_1-p_2|}\left(\frac{\nu^3}{2} + \left(\nu + \frac{\nu^2}{4}\right)\mu\right) = \varepsilon_{\lambda^*}$$

To obtain the regret bound on the empirical rates, we apply assumptions (ii), (iii) once again to bound the expected difference between $\lambda^*(p)$ and $\frac{1}{T}|S_u(p;T)|$, and apply the triangle inequality:

$$\mathbb{E}_{u,\pi}\left[\left|\frac{1}{T}|S_u(p^*;T)| - \frac{1}{T}|S_u(\hat{p};T)|\right|\right] \leq \varepsilon_{\lambda^*} + 2(\varepsilon_{\text{dev}} + \varepsilon_{\text{LV}})$$

Note that $\frac{\nu}{|p_1-p_2|} > 1$, as $\frac{\beta}{\alpha} \geq 1$ since all the users are assumed to remain engaged in the long term, and $|p_1 - p_2| \leq 1$ as $p_1, p_2 \in [0,1]$. Therefore, the function $\eta_{\text{TPP}} = g(\mu, \nu) = \left(\frac{\nu^3}{2} + \left(\nu + \frac{\nu^2}{4}\right)\eta + 2\nu\right)$ satisfies:

$$\mathbb{E}_{u,\pi}\left[\left|\frac{1}{T}|S_u(p^*;T)| - \frac{1}{T}|S_u(\hat{p};T)|\right|\right] \leq \frac{\eta_{\text{TPP}}}{|p_1-p_2|}(\varepsilon_{\text{pred}} + \varepsilon_{\text{dev}} + \varepsilon_{\text{LV}})$$

$\square$

# B    Experimental details

## B.1    Data

We base our experimental environment on the MovieLens 1M dataset, which is a standard benchmark dataset used widely in recommendation system research (Harper and Konstan, 2015). The dataset includes 1,000,209 ratings provided by 6,040 users and for 3,706 movies. Rating are in the range $\{1, \ldots, 5\}$, and all users in the dataset have at least 20 reported ratings. The dataset is publicly available at: https://grouplens.org/datasets/movielens/1m/.

**Data partitioning.**    To learn latent user and item features, 30% of all ratings were drawn at random. Stratified sampling was applied to ensure that all users and items were covered, and so that each users have roughly the same proportion of ratings used for this step. These ratings were only used only for learning a CF model, and were discarded afterwards. The remaining 70% data points were used for training and testing. For these, we first randomly sampled 1,000 users to form the test set. Then, the remaining users were partitioned into the main train set $\mathcal{S}$, which included 70% ($\approx$3,528) of these users, and the experimental treatment sets $\mathcal{S}^{(j)}$, each including 10% ($\approx$504) users for $N = 3$. This procedure was repeated 10 times, and we report average results and standard errors.

## B.2    Implementation details

- **Hardware**: All experiments were run on a single laptop, with 16GB of RAM, M1 Pro processor, and with no GPU support.
- **Runtime**: A single run consisting the entire pipeline (data loading and partitioning, collaborative filtering, training classifiers, simulating dynamics, learning policies, measuring and comparing performance) takes roughly 20 minutes. The main bottleneck is the discrete LV simulation, taking roughly 70% of runtime to compute, mostly due to bookkeeping necessary for the non-stationary baselines. Simulation code was optimized using the NUMBA jit compiler, which improves runtime.
- **Optimization packages**:
  - **Collaborative filtering (CF)**: We use the SURPRISE package (Hug, 2020), which includes an implementation of the SVD algorithm for CF. All parameters were set to default values.
  - **Regression**: We use the SCIKIT-LEARN implementation of linear regression for predicting long-term engagement from user features (i.e the prediction models $f_j(u)$ in Eq. (8)). All parameters were set to default values.
  - **Non-Negative Least Squares (NNLS)**: We use the SCIPY.OPTIMIZE implementation of NNLS. The algorithm was used with its default parameters.
- **Code**: Code for reproducing all of our figures and experiments is available in the following repository: https://github.com/edensaig/take-a-break.

## B.3    Other baselines

- **Safety**: In each step of the TPP simulation, look $k$ step back, and calculate the empirical rate $\tilde{\lambda}_i = \frac{k}{t_i - t_{i-k}}$. If this rate exceeds the threshold $\tilde{\lambda}_i > \tau$, the policy enters a 'cool-down' policy state, serving only forced breaks until the next time period. In our experiments, we used thresholds $\tau \in \{14, 16\}$, $k = 10$ look-behind steps, and defined the cool-down period as 0.5 time units.

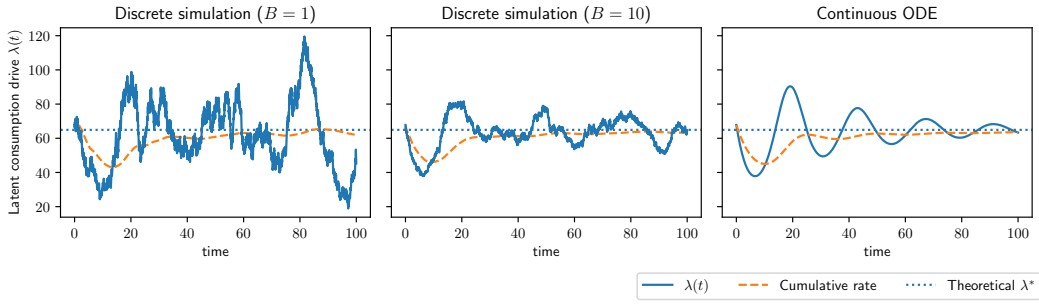

Figure 5: Example discrete sequence $S_u \sim \text{TPP}(u; T)$, compared to continuous LV dynamics. Our TPP produces discrete sequences that are qualitatively different from their continuous-time analogs (blue lines), Nonetheless, it captures the general properties of our proposed behavioral model: note how cumulative averaging behavior (orange dashes) exhibits 'habit formation', which our equilibrium approach targets (blue dots). For the same initial conditions $\lambda(0), q(0)$, the figure shows how varying the number of recommended items per step ($B$) 'smooths' the discrete behavior (left: $B = 1$, center: $B = 10$). For fixed $\beta_u(t) = \beta_u$, when $B \to \infty$, and when $\Delta t \to 0$, TPP sequences approach a continuous LV trajectory; in general, and particularly when $\beta_u(t)$ varies by step and per recommended items—this is not the case.

- **Oracle**: To estimate the effect of perfect predictions, we implement an oracle predictor $f_p^{\text{oracle}}(u)$ which has access to the latent user parameters. For a given $u$ and for each $p$, the predictor outputs the infinite-horizon LV equilibrium for $u$, namely $f_p^{\text{oracle}}(u) = \lambda^*(p; \tilde{\theta}_u)$. We define $\tilde{\theta}_u = (\alpha_u, \tilde{\beta}_u, \gamma_u, \delta_u)$, where $\alpha_u, \gamma_u, \delta_u$ are the unobserved parameters for the given user, and $\tilde{\beta}_u$ is the expected value of $\beta_{ux}$ induced by the distribution over recommended items $x$ induced by the recommendation policy $\psi$. We view $\tilde{\theta}_u$ as a useful proxy for the otherwise unattainable $\bar{\theta}_u$.

### B.4 Hyperparameters

- **Collaborative filtering**: We used $d = 8$ latent factors and enabled bias terms, which ensured performance is close to the benchmark of $\text{RMSE} = 0.873$ reported in the SURPRISE documentation. We used the vanilla SVD solver, with all hyper-parameters set to their default values.

- **Recommendation policy**: Softmax temperature was set to 0.5.

- **Prediction**: We trained regressors $f(u)$ on input feature vectors $u \in \mathbb{R}^{d+2}$ consisting of: (i) SVD latent user factors, (ii) SVD user bias terms, (iii) an additional feature consisting of the average predicted ratings for unseen items (exponentiated and normalized), which we found to slightly improve predictive performance. We chose to focus on linear models since the treatment datasets are relatively small (each $|\mathcal{S}^{(j)}| \approx 500$), and since other model classes (including boosted trees and MLPs) did not perform significantly better.

- **Discrete TPP**: Interaction sequences for each user were generated according to an LV discretization scheme, described in detail in the next section. Latent sates were initialized randomly with relative uniform noise around the theoretical LV equilibrium point $(\lambda_0, q_0) = ((1 + \xi_\lambda)\lambda^*, (1 + \xi_q)q^*)$, where $\xi_\lambda, \xi_q \sim \text{Uniform}(-0.1, 0.1)$. Latent states were updated each $B = 10$ recommendations to stabilize noise (see Figure 5). When $x$ is recommended to $u$ at time $t$, latent states and $\Delta t$ are set according to $\beta_u(t)$, which depends on ratings $r_{ux}$ (true or mixed with predictions $u^\top x$ via $\kappa$). Specifically, we use $\beta_u(t) = r_{ux}^2/5 \in \{0.2, 0.8, 1.8, 3.2, 5\}$, which is convex, to accentuate the role of low ratings since they are underrepresented in the data. For $B \geq 1$, we take the effective $\beta_u(t)$ to be the average over the $B$ items recommended in that step. We set $\alpha = 1.3$, and chose $\gamma = 0.2, \delta = 0.01$ (which together determine scale) so that typical values for engagement rate $\frac{1}{T}|S_u|$ are on the order of $\approx 10$ for the chosen $T = 100$.

### B.5 Discrete TPP

The TPP we use for simulating user behavior is based on a discretization of the LV system described in Eq. (2), based on the forward Euler method with variable step sizes.

Each user is associated with discrete latent states $\lambda_i, q_i$, and parameters $\alpha_u, \gamma_u, \delta_u$. Initial states $\lambda_0, q_0$ are set randomly. At each step, and in time $t_i$, the system recommends $x_i = x(t_i)$, which triggers updates in latent states, and determines the next time of interaction $t_{i+1}$. As noted, these update depend on item-specific parameters $\beta_{u,x_i}$.

Under stationary policy $\pi(p)$, the system recommends an item with probability $(1 - p)$, and suggests a break with probability $p$. The simulator considers $B$ recommendation opportunities at each step. For each $k \in \{1, \ldots, B\}$, denote by $I_k \in \{0, 1\}$ the break indicator, equal to 0 when a break is recommended at the $k$-th slot in the batch. Denote by $x \sim \psi$ the item recommended by the underlying policy $\psi$, and by $\beta(x) = r_{ux}^2/5$ the corresponding LV hyperparameter as defined above. For a given horizon $T$, the TPP process generating $S_u$ is described by Algorithm 1:

---

**Algorithm 1:** Discrete TPP for user $u$

---

**Input:** Break probability $p \in [0, 1]$
           Stationary content recommendation probability $\psi$
           Time horizon $T > 0$
**Output:** Interaction sequence $S_u \sim \text{TPP}_{\pi(p) \circ \psi}(u; T)$
Initialize $i = 0$, $t_0 = 0$, $S_u = \{\}$;
**while** $t_i < T$ **do**
    **foreach** $k \in \{1, \ldots, B\}$ **do**
        $I_k \sim \text{Bernoulli}(1 - p)$;
        $x_k \sim \psi$;
        $\beta_k \longleftarrow \beta(r_{ux_k})$;
    **end**
    $\Delta t_i = \lambda_i^{-1}$ ;
    $\lambda_{i+1} \longleftarrow \lambda_i + \left( -\alpha + \frac{\sum_{k=1}^{B} I_k \beta_k}{B} q_i \right) \lambda_i \Delta t_i$ ;
    $q_{i+1} \longleftarrow q_i + \left( \gamma(1 - q_i) - \frac{\sum_{k=1}^{B} I_k \delta}{B} \lambda_i \right) q_i \Delta t_i$ ;
    $t_{i+1} \longleftarrow t_i + \lambda_{i+1}^{-1}$ ;
    $S_u \longleftarrow S_u \cup \{(t_i, (x_1, \ldots, x_B), (I_1, \ldots, I_B))\}$ ;
    $i \longleftarrow i + 1$ ;
**end**

---

