# OpenReview forum: "Learning to Take a Break: Sustainable Optimization of Long-Term User Engagement"
_NeurIPS.cc/2022/Workshop/TSRML — TSRML2022_

### Official Review · Reviewer_aS5S · 2022-10-14
**Interesting idea with solid theoretical analysis, but the relation to trustworthiness is unclear**

**Overall Rating:** 6

**Summary:**

This paper proposes an interesting idea of learning a break policy to let the user more engaged in a recommendation system. The proposed behavior model is formulated with the Lotka-Volterra dynamics. The results on semi-synthetic dataset show the advantage of the method.

**Strengths:**

The proposed problem setting is interesting for a recommendation system. The theory is sound and the experiments are solid.

**Weaknesses:**

The major weakness is that the relation to the trustworthy ML workshop is unclear, because I could not find any explanation or discussions about why the proposed method is trustworthy, such as robust, fair, or private.

**Overall Recommendation:**

The proposed method is solid in solving an important problem in recommendation system, but it seems that the topic is not aligned with the workshop's topic very well.

**Review Confidence:**

3: The reviewer is fairly confident that the evaluation is correct

---

### Official Review · Reviewer_4ue7 · 2022-10-20
**Interesting paper**

**Overall Rating:** 6

**Summary:**

In this paper, the authors argue that Lokta-Volterra dynamics can model users acting based on two balancing latent states called "drive" and "interest". They further explain that the value of $\lambda(t)$ can be influenced by a control input $p(t)$. The idea is to promote digital wellbeing by learning "breaking" policies that prompt the user to take a break (where the breaking policy is $p(t)$). Experiments are conducted on semi-synthetic data.



**Strengths:**

* Promoting digital wellbeing is an important subject.
* Modeling user dynamics using the Lotka-Volterra dynamics is interesting (I don't think I have seen this particular formulation in that context before).
* The formalization seems rigorous and correct and the paper is well-written.

**Weaknesses:**

* The workshop clearly recommends 6 pages. While 9+ pages is acceptable, I believe that the paper can be drastically shortened. The related work (while relevant) can be moved to the appendix. The introduction can be shortened.
* The paper fails to demonstrate why Lotka-Volterra dynamics are representative of actual users. The experiments look at a single dataset (MovieLens 1M). This dataset does not contain information about user taking breaks (as far as I can tell).
* Overall, the paper reads as a toy experiment without clear grounding to user studies or real recommender system.

I would strongly encourage the authors to look at more datasets and try to assess whether Lotka-Volterra dynamics (with $p(t)$ as a breaking policy) is representative of the real world.

**Overall Recommendation:**

Overall, while I am concerned about the faithfulness of the model, I did enjoy reading the paper and find it a valuable contribution to the workshop. I believe the paper could be strengthened by shortening it and really focusing on the key questions: (1) Are Lotka-Volterra dynamics representative of the real world? (2) Can a breaking policy be modeled using $p(t)$? (3) Do conclusions generalize beyond the MovieLens 1M dataset?

**Review Confidence:**

1: The reviewer's evaluation is an educated guess

---

### Decision · Program_Chairs · 2022-10-23

**Decision:**

Accept

**Comment:**

Following the unanimous recommendations from reviewers, the submission is accepted.